# Sparse PCA via Covariance Thresholding

**Yash Deshpande**
Electrical Engineering
Stanford University
yashd@stanford.edu

**Andrea Montanari**
Electrical Engineering and Statistics
Stanford University
montanari@stanford.edu

## Abstract

In sparse principal component analysis we are given noisy observations of a low-rank matrix of dimension $n \times p$ and seek to reconstruct it under additional sparsity assumptions. In particular, we assume here that the principal components $\mathbf{v}_1, \ldots, \mathbf{v}_r$ have at most $k_1, \cdots, k_q$ non-zero entries respectively, and study the high-dimensional regime in which $p$ is of the same order as $n$.

In an influential paper, Johnstone and Lu [JL04] introduced a simple algorithm that estimates the support of the principal vectors $\mathbf{v}_1, \ldots, \mathbf{v}_r$ by the largest entries in the diagonal of the empirical covariance. This method can be shown to succeed with high probability if $k_q \le C_1 \sqrt{n/\log p}$, and to fail with high probability if $k_q \ge C_2 \sqrt{n/\log p}$ for two constants $0 < C_1, C_2 < \infty$. Despite a considerable amount of work over the last ten years, no practical algorithm exists with provably better support recovery guarantees.

Here we analyze a covariance thresholding algorithm that was recently proposed by Krauthgamer, Nadler and Vilenchik [KNV13]. We confirm empirical evidence presented by these authors and rigorously prove that the algorithm succeeds with high probability for $k$ of order $\sqrt{n}$. Recent conditional lower bounds [BR13] suggest that it might be impossible to do significantly better.

The key technical component of our analysis develops new bounds on the norm of kernel random matrices, in regimes that were not considered before.

## 1 Introduction

In the spiked covariance model proposed by [JL04], we are given data $\mathbf{x}_1, \mathbf{x}_2, \ldots, \mathbf{x}_n$ with $\mathbf{x}_i \in \mathbb{R}^p$ of the form[1]:

$$\mathbf{x}_i = \sum_{q=1}^{r} \sqrt{\beta_q}\, u_{q,i}\, \mathbf{v}_q + \mathbf{z}_i \,, \tag{1}$$

Here $\mathbf{v}_1, \ldots, \mathbf{v}_r \in \mathbb{R}^p$ is a set of orthonormal vectors, that we want to estimate, while $u_{q,i} \sim \mathsf{N}(0,1)$ and $\mathbf{z}_i \sim \mathsf{N}(0, \mathrm{I}_p)$ are independent and identically distributed. The quantity $\beta_q \in \mathbb{R}_{>0}$ quantifies the signal-to-noise ratio. We are interested in the high-dimensional limit $n, p \to \infty$ with $\lim_{n \to \infty} p/n = \alpha \in (0, \infty)$. In the rest of this introduction we will refer to the rank one case, in order to simplify the exposition, and drop the subscript $q = \{1, 2, \ldots, r\}$. Our results and proofs hold for general bounded rank.

The standard method of principal component analysis involves computing the sample covariance matrix $\mathbf{G} = n^{-1} \sum_{i=1}^{n} \mathbf{x}_i \mathbf{x}_i^\mathsf{T}$ and estimates $\mathbf{v} = \mathbf{v}_1$ by its principal eigenvector $\mathbf{v}_{\mathrm{PC}}(\mathbf{G})$. It is a well-known fact that, in the high dimensional asymptotic $p/n \to \alpha > 0$, this yields an inconsistent

estimate [JL09]. Namely $\|\mathbf{v}_{\mathrm{PC}} - \mathbf{v}\|_2 \not\to 0$ in the high-dimensional asymptotic limit, unless $\alpha \to 0$ (i.e. $p/n \to 0$). Even worse, Baik, Ben-Arous and Péché [BBAP05] and Paul [Pau07] demonstrate a phase transition phenomenon: if $\beta < \sqrt{\alpha}$ the estimate is asymptotically orthogonal to the signal $\langle \mathbf{v}_{\mathrm{PC}}, \mathbf{v} \rangle \to 0$. On the other hand, for $\beta > \sqrt{\alpha}$, $\langle \mathbf{v}_{\mathrm{PC}}, \mathbf{v} \rangle$ remains strictly positive as $n, p \to \infty$. This phase transition phenomenon has attracted considerable attention recently within random matrix theory [FP07, CDMF09, BGN11, KY13].

These inconsistency results motivated several efforts to exploit additional structural information on the signal $\mathbf{v}$. In two influential papers, Johnstone and Lu [JL04, JL09] considered the case of a signal $\mathbf{v}$ that is sparse in a suitable basis, e.g. in the wavelet domain. Without loss of generality, we will assume here that $\mathbf{v}$ is sparse in the canonical basis $\mathbf{e}_1, \ldots \mathbf{e}_p$. In a nutshell, [JL09] proposes the following:

1. Order the diagonal entries of the Gram matrix $\mathbf{G}_{i(1),i(1)} \geq \mathbf{G}_{i(2),i(2)} \geq \cdots \geq \mathbf{G}_{i(p),i(p)}$, and let $J \equiv \{i(1), i(2), \ldots, i(k)\}$ be the set of indices corresponding to the $k$ largest entries.

2. Set to zero all the entries $\mathbf{G}_{i,j}$ of $\mathbf{G}$ unless $i, j \in J$, and estimate $\mathbf{v}$ with the principal eigenvector of the resulting matrix.

Johnstone and Lu formalized the sparsity assumption by requiring that $\mathbf{v}$ belongs to a weak $\ell_q$-ball with $q \in (0, 1)$. Instead, here we consider a strict sparsity constraint where $\mathbf{v}$ has exactly $k$ non-zero entries, with magnitudes bounded below by $\theta/\sqrt{k}$ for some constant $\theta > 0$. The case of $\theta = 1$ was studied by Amini and Wainwright in [AW09].

Within this model, it was proved that diagonal thresholding successfully recovers the support of $\mathbf{v}$ provided $\mathbf{v}$ is sparse enough, namely $k \leq C\sqrt{n/\log p}$ with $C = C(\alpha, \beta)$ a constant [AW09]. (Throughout the paper we denote by $C$ constants that can change from point to point.) This result is a striking improvement over vanilla PCA. While the latter requires a number of samples scaling as the number of parameters[2] $n \gtrsim p$, sparse PCA via diagonal thresholding achieves the same objective with a number of samples scaling as the number of *non-zero* parameters, $n \gtrsim k^2 \log p$.

At the same time, this result is not as optimistic as might have been expected. By searching exhaustively over all possible supports of size $k$ (a method that has complexity of order $p^k$) the correct support can be identified with high probability as soon as $n \gtrsim k \log p$. On the other hand, no method can succeed for much smaller $n$, because of information theoretic obstructions [AW09].

Over the last ten years, a significant effort has been devoted to developing practical algorithms that outperform diagonal thresholding, see e.g. [MWA05, ZHT06, dEGJL07, dBG08, WTH09]. In particular, d'Aspremont et al. [dEGJL07] developed a promising M-estimator based on a semidefinite programming (SDP) relaxation. Amini and Wainwright [AW09] carried out an analysis of this method and proved that, if *(i)* $k \leq C(\beta) \, n/\log p$, and *(ii)* if the SDP solution has rank one, then the SDP relaxation provides a consistent estimator of the support of $\mathbf{v}$.

At first sight, this appears as a satisfactory solution of the original problem. No procedure can estimate the support of $\mathbf{v}$ from less than $k \log p$ samples, and the SDP relaxation succeeds in doing it from –at most– a constant factor more samples. This picture was upset by a recent, remarkable result by Krauthgamer, Nadler and Vilenchik [KNV13] who showed that the rank-one condition assumed by Amini and Wainwright does not hold for $\sqrt{n} \lesssim k \lesssim (n/\log p)$. This result is consistent with recent work of Berthet and Rigollet [BR13] demonstrating that sparse PCA cannot be performed in polynomial time in the regime $k \gtrsim \sqrt{n}$, under a certain computational complexity conjecture for the so-called planted clique problem.

In summary, the problem of support recovery in sparse PCA demonstrates a fascinating interplay between computational and statistical barriers.

**From a statistical perspective,** and disregarding computational considerations, the support of $\mathbf{v}$ can be estimated consistently if and only if $k \lesssim n/\log p$. This can be done, for instance, by exhaustive search over all the $\binom{p}{k}$ possible supports of $\mathbf{v}$. (See [VL12, CMW$^+$13] for a minimax analysis.)

**From a computational perspective,** the problem appears to be much more difficult. There is rigorous evidence [BR13, MW13] that no polynomial algorithm can reconstruct the support unless $k \lesssim \sqrt{n}$. On the positive side, a very simple algorithm (Johnstone and Lu's diagonal thresholding) succeeds for $k \lesssim \sqrt{n/\log p}$.

Of course, several elements are still missing in this emerging picture. In the present paper we address one of them, providing an answer to the following question:

> *Is there a polynomial time algorithm that is guaranteed to solve the sparse PCA problem with high probability for $\sqrt{n/\log p} \lesssim k \lesssim \sqrt{n}$?*

We answer this question positively by analyzing a covariance thresholding algorithm that proceeds, briefly, as follows. (A precise, general definition, with some technical changes is given in the next section.)

1. Form the Gram matrix $\mathbf{G}$ and set to zero all its entries that are in modulus smaller than $\tau/\sqrt{n}$, for $\tau$ a suitably chosen constant.
2. Compute the principal eigenvector $\widehat{\mathbf{v}}_1$ of this thresholded matrix.
3. Denote by $\mathsf{B} \subseteq \{1, \ldots, p\}$ be the set of indices corresponding to the $k$ largest entries of $\widehat{\mathbf{v}}_1$.
4. Estimate the support of $\mathbf{v}$ by 'cleaning' the set $\mathsf{B}$. (Briefly, $\mathbf{v}$ is estimated by thresholding $\mathbf{G}\widehat{\mathbf{v}}_{\mathsf{B}}$ with $\widehat{\mathbf{v}}_{\mathsf{B}}$ obtained by zeroing the entries outside $\mathsf{B}$.)

Such a covariance thresholding approach was proposed in [KNV13], and is in turn related to earlier work by Bickel and Levina [BL08]. The formulation discussed in the next section presents some technical differences that have been introduced to simplify the analysis. Notice that, to simplify proofs, we assume $k$ to be known: This issue is discussed in the next two sections.

The rest of the paper is organized as follows. In the next section we provide a detailed description of the algorithm and state our main results. Our theoretical results assume full knowledge of problem parameters for ease of proof. In light of this, in Section 3 we discuss a practical implementation of the same idea that does not require prior knowledge of problem parameters, and is entirely data-driven. We also illustrate the method through simulations. The complete proofs are available in the accompanying supplement, in Sections 1, 2 and 3 respectively.

## 2 Algorithm and main result

For notational convenience, we shall assume hereafter that $2n$ sample vectors are given (instead of $n$): $\{\mathbf{x}_i\}_{1 \leq i \leq 2n}$. These are distributed according to the model (1). The number of spikes $r$ will be treated as a known parameter in the problem.

We will make the following assumptions:

**A1** The number of spikes $r$ and the signal strengths $\beta_1, \ldots, \beta_r$ are fixed as $n, p \to \infty$. Further $\beta_1 > \beta_2 > \ldots \beta_r$ are all *distinct*.

**A2** Let $\mathsf{Q}_q$ and $k_q$ denote the support of $\mathbf{v}_q$ and its size respectively. We let $\mathsf{Q} = \cup_q \mathsf{Q}_q$ and $k = \sum_q k_q$ throughout. Then the non-zero entries of the spikes satisfy $|v_{q,i}| \geq \theta/\sqrt{k_q}$ for all $i \in \mathsf{Q}_q$ for some $\theta > 0$. Further, for any $q, q'$ we assume $|v_{q,i}/v_{q',i}| \leq \gamma$ for every $i \in \mathsf{Q}_q \cap \mathsf{Q}_{q'}$, for some constant $\gamma > 1$.

As before, we are interested in the high-dimensional limit of $n, p \to \infty$ with $p/n \to \alpha$. A more detailed description of the covariance thresholding algorithm for the general model (1) is given in Algorithm 1. We describe the basic intuition for the simpler rank-one case (omitting the subscript $q \in \{1, 2, \ldots, r\}$), while stating results in generality.

We start by splitting the data into two halves: $(\mathbf{x}_i)_{1 \leq i \leq n}$ and $(\mathbf{x}_i)_{n < i \leq 2n}$ and compute the respective sample covariance matrices $\mathbf{G}$ and $\mathbf{G}'$ respectively. As we will see, the matrix $\mathbf{G}$ is used to obtain a good estimate for the spike $\mathbf{v}$. This estimate, along with the (independent) second part $\mathbf{G}'$, is then used to construct a consistent estimator for the supports of $\mathbf{v}$.

Let us focus on the first phase of the algorithm, which aims to obtain a good estimate of $\mathbf{v}$. We first compute $\widehat{\boldsymbol{\Sigma}} = \mathbf{G} - \mathrm{I}$. For $\beta > \sqrt{\alpha}$, the principal eigenvector of $\mathbf{G}$, and hence of $\widehat{\boldsymbol{\Sigma}}$ is positively correlated with $\mathbf{v}$, i.e. $\lim_{n\to\infty}\langle\widehat{\mathbf{v}}_1(\widehat{\boldsymbol{\Sigma}}), \mathbf{v}\rangle > 0$. However, for $\beta < \sqrt{\alpha}$, the noise component in $\widehat{\boldsymbol{\Sigma}}$ dominates and the two vectors become asymptotically orthogonal, i.e. for instance $\lim_{n\to\infty}\langle\widehat{\mathbf{v}}_1(\widehat{\boldsymbol{\Sigma}}), \mathbf{v}\rangle = 0$. In order to reduce the noise level, we exploit the sparsity of the spike $\mathbf{v}$.

Denoting by $\mathbf{X} \in \mathbb{R}^{n\times p}$ the matrix with rows $\mathbf{x}_1, \ldots \mathbf{x}_n$, by $\mathbf{Z} \in \mathbb{R}^{n\times p}$ the matrix with rows $\mathbf{z}_1, \ldots \mathbf{z}_n$, and letting $\mathbf{u} = (u_1, u_2, \ldots, u_n)$, the model (1) can be rewritten as

$$\mathbf{X} = \sqrt{\beta}\,\mathbf{u}\,\mathbf{v}^{\mathsf{T}} + \mathbf{Z}\,. \tag{2}$$

Hence, letting $\beta' \equiv \beta\|u\|^2/n \approx \beta$, and $\mathbf{w} \equiv \sqrt{\beta}\mathbf{Z}^{\mathsf{T}}\mathbf{u}/n$

$$\widehat{\boldsymbol{\Sigma}} = \beta'\,\mathbf{v}\mathbf{v}^{\mathsf{T}} + \mathbf{v}\,\mathbf{w}^{\mathsf{T}} + \mathbf{w}\,\mathbf{v}^{\mathsf{T}} + \frac{1}{n}\mathbf{Z}^{\mathsf{T}}\mathbf{Z} - \mathrm{I}_p,. \tag{3}$$

For a moment, let us neglect the cross terms $(\mathbf{v}\mathbf{w}^{\mathsf{T}} + \mathbf{w}\mathbf{v}^{\mathsf{T}})$. The 'signal' component $\beta'\,\mathbf{v}\mathbf{v}^{\mathsf{T}}$ is sparse with $k^2$ entries of magnitude $\beta/k$, which (in the regime of interest $k = \sqrt{n}/C$) is equivalent to $C\beta/\sqrt{n}$. The 'noise' component $\mathbf{Z}^{\mathsf{T}}\mathbf{Z}/n - \mathrm{I}_p$ is dense with entries of order $1/\sqrt{n}$. Assuming $k/\sqrt{n}$ a small enough constant, it should be possible to remove most of the noise by thresholding the entries at level of order $1/\sqrt{n}$. For technical reasons, we use the soft thresholding function $\eta : \mathbb{R} \times \mathbb{R}_{\geq 0} \to \mathbb{R}$, $\eta(z;\tau) = \mathrm{sgn}(z)(|z| - \tau)_+$. We will omit the second argument wherever it is clear from context. Classical denoising theory [DJ94, Joh02] provides upper bounds the estimation error of such a procedure. Note however that these results fall short of our goal. Classical theory measures estimation error by (element-wise) $\ell_p$ norm, while here we are interested in the resulting principal eigenvector. This would require bounding, for instance, the error in operator norm.

Since the soft thresholding function $\eta(z;\tau/\sqrt{n})$ is affine when $z \gg \tau/\sqrt{n}$, we would expect that the following decomposition holds approximately (for instance, in operator norm):

$$\eta(\widehat{\boldsymbol{\Sigma}}) \approx \eta\left(\beta'\mathbf{v}\mathbf{v}^{\mathsf{T}}\right) + \eta\left(\frac{1}{n}\mathbf{Z}^{\mathsf{T}}\mathbf{Z} - \mathrm{I}_p\right). \tag{4}$$

The main technical challenge now is to control the operator norm of the perturbation $\eta(\mathbf{Z}^{\mathsf{T}}\mathbf{Z}/n - \mathrm{I}_p)$. It is easy to see that $\eta(\mathbf{Z}^{\mathsf{T}}\mathbf{Z}/n - \mathrm{I}_p)$ has entries of variance $\delta(\tau)/n$, for $\delta(\tau) \to 0$ as $\tau \to \infty$. If entries were independent with mild decay, this would imply –with high probability–

$$\left\|\eta\left(\frac{1}{n}\mathbf{Z}^{\mathsf{T}}\mathbf{Z}\right)\right\|_2 \lesssim C\delta(\tau), \tag{5}$$

for some constant $C$. Further, the first component in the decomposition (4) is still approximately rank one with norm of the order of $\beta' \approx \beta$. Consequently, with standard linear algebra results on the perturbation of eigenspaces [DK70], we obtain an error bound $\|\widehat{\mathbf{v}} - \mathbf{v}\| \lesssim \delta(\tau)/C'\beta$

Our first theorem formalizes this intuition and provides a bound on the estimation error in the principal components of $\eta(\widehat{\boldsymbol{\Sigma}})$.

**Theorem 1.** *Under the spiked covariance model Eq.* (1) *satisfying Assumption* A1, *let $\widehat{\mathbf{v}}_q$ denote the $q^{th}$ eigenvector of $\eta(\widehat{\boldsymbol{\Sigma}})$ using threshold $\tau$. For every $\alpha, (\beta_q)_{q=1}^r \in (0,\infty)$, integer $r$ and every $\varepsilon > 0$ there exist constants, $\tau = \tau(\varepsilon, \alpha, (\beta_q)_{q=1}^r, r, \theta)$ and $0 < c_* = c_*(\varepsilon, \alpha, (\beta_q)_{q=1}^r, r, \theta) < \infty$ such that, if $\sum_q k_q = \sum_q |\mathrm{supp}(\mathbf{v}_q)| \leq c_*\sqrt{n}$), then*

$$\mathbb{P}\left\{ \min(\|\widehat{\mathbf{v}}_q - \mathbf{v}_q\|, \|\widehat{\mathbf{v}}_q + \mathbf{v}_q\|) \leq \varepsilon \ \forall q \in \{1, \ldots, r\} \right\} \geq 1 - \frac{\alpha}{n^4}. \tag{6}$$

Random matrices of the type $\eta(\mathbf{Z}^{\mathsf{T}}\mathbf{Z}/n - \mathrm{I}_p)$ are called inner-product kernel random matrices and have attracted recent interest within probability theory [EK10a, EK10b, CS12]. In this literature, the asymptotic eigenvalue distribution of a matrix $f(\mathbf{Z}^{\mathsf{T}}\mathbf{Z}/n)$ is the object of study. Here $f : \mathbb{R} \to \mathbb{R}$ is a kernel function and is applied entry-wise to the matrix $\mathbf{Z}^{\mathsf{T}}\mathbf{Z}/n$, with $\mathbf{Z}$ a matrix as above. Unfortunately, these results cannot be applied to our problem for the following reasons:

- The results [EK10a, EK10b] are perturbative and provide conditions under which the spectrum of $f(\mathbf{Z}^{\mathsf{T}}\mathbf{Z}/n)$ is close to a rescaling of the spectrum of $(\mathbf{Z}^{\mathsf{T}}\mathbf{Z}/n)$ (with rescaling

---

**Algorithm 1** Covariance Thresholding

1: **Input:** Data $(\mathbf{x}_i)_{1 \leq i \leq 2n}$, parameters $r, (k_q)_{q \leq r} \in \mathbb{N}, \theta, \tau, \rho \in \mathbb{R}_{\geq 0}$;
2: Compute the Gram matrices $\mathbf{G} \equiv \sum_{i=1}^{n} \mathbf{x}_i \mathbf{x}_i^{\mathsf{T}}/n$, $\mathbf{G}' \equiv \sum_{i=n+1}^{2n} \mathbf{x}_i \mathbf{x}_i^{\mathsf{T}}/n$;
3: Compute $\widehat{\boldsymbol{\Sigma}} = \mathbf{G} - \mathrm{I}_p$ (resp. $\widehat{\boldsymbol{\Sigma}}' = \mathbf{G}' - \mathrm{I}_p$);
4: Compute the matrix $\eta(\widehat{\boldsymbol{\Sigma}})$ by soft-thresholding the entries of $\widehat{\boldsymbol{\Sigma}}$:

$$\eta(\widehat{\boldsymbol{\Sigma}})_{ij} = \begin{cases} \widehat{\boldsymbol{\Sigma}}_{ij} - \frac{\tau}{\sqrt{n}} & \text{if } \widehat{\boldsymbol{\Sigma}}_{ij} \geq \tau/\sqrt{n}, \\ 0 & \text{if } -\tau/\sqrt{n} < \widehat{\boldsymbol{\Sigma}}_{ij} < \tau/\sqrt{n}, \\ \widehat{\boldsymbol{\Sigma}}_{ij} + \frac{\tau}{\sqrt{n}} & \text{if } \widehat{\boldsymbol{\Sigma}}_{ij} \leq -\tau/\sqrt{n}, \end{cases}$$

5: Let $(\widehat{\mathbf{v}}_q)_{q \leq r}$ be the first $r$ eigenvectors of $\eta(\widehat{\boldsymbol{\Sigma}})$;
6: Define $\mathbf{s}_q \in \mathbb{R}^p$ by $s_{q,i} = \widehat{v}_{q,i} \mathbb{I}(|\widehat{v}_{q,i} \geq \theta/2\sqrt{k_q}|)$;
7: **Output:** $\widehat{\mathsf{Q}} = \{i \in [p] : \exists q \text{ s.t. } |(\widehat{\boldsymbol{\Sigma}}' \mathbf{s}_q)_i| \geq \rho\}$.

---

factors depending on the Taylor expansion of $f$ close to 0). We are interested instead in a non-perturbative regime in which the spectrum of $f(\mathbf{Z}^{\mathsf{T}}\mathbf{Z}/n)$ is very different from the one of $(\mathbf{Z}^{\mathsf{T}}\mathbf{Z}/n)$ (and the Taylor expansion is trivial).

- [CS12] consider $n$-dependent kernels, but their results are asymptotic and concern the weak limit of the empirical spectral distribution of $f(\mathbf{Z}^{\mathsf{T}}\mathbf{Z}/n)$. This does not yield an upper bound on the spectral norm[3] of $f(\mathbf{Z}^{\mathsf{T}}\mathbf{Z}/n)$.

Our approach to prove Theorem 1 follows instead the so-called $\varepsilon$-net method: we develop high probability bounds on the maximum Rayleigh quotient:

$$\max_{\mathbf{y} \in S^{p-1}} \langle \mathbf{y}, \eta(\mathbf{Z}^{\mathsf{T}}\mathbf{Z}/n)\mathbf{y} \rangle = \max_{\mathbf{y} \in S^{p-1}} \sum_{i,j} \eta\left(\frac{\langle \tilde{\mathbf{z}}_i, \tilde{\mathbf{z}}_j \rangle}{n}; \frac{\tau}{\sqrt{n}}\right) y_i y_j,$$

Here $S^{p-1} = \{\mathbf{y} \in \mathbb{R}^p : \|\mathbf{y}\| = 1\}$ is the unit sphere and $\tilde{\mathbf{z}}_i$ denote the columns of $\mathbf{Z}$. Since $\eta(\mathbf{Z}^{\mathsf{T}}\mathbf{Z}/n)$ is not Lipschitz continuous in the underlying Gaussian variables $\mathbf{Z}$, concentration does not follow immediately from Gaussian isoperimetry. We have to develop more careful (non-uniform) bounds on the gradient of $\eta(\mathbf{Z}^{\mathsf{T}}\mathbf{Z}/n)$ and show that they imply concentration as required.

While Theorem 1 guarantees that $\widehat{\mathbf{v}}$ is a reasonable estimate of the spike $\mathbf{v}$ in $\ell_2$ distance (up to a sign flip), it does not provide a consistent estimator of its support. This brings us to the second phase of our algorithm. Although $\widehat{\mathbf{v}}$ is not even expected to be sparse, it is easy to see that the largest entries of $\widehat{\mathbf{v}}$ should have significant overlap with $\mathrm{supp}(\mathbf{v})$. Steps 6, 7 and 8 of the algorithm exploit this property to construct a consistent estimator $\widehat{\mathsf{Q}}$ of the support of the spike $\mathbf{v}$. Our second theorem guarantees that this estimator is indeed consistent.

**Theorem 2.** *Consider the spiked covariance model of Eq.* (1) *satisfying Assumptions* A1, A2. *For any $\alpha, (\beta_q)_{q \leq r} \in (0, \infty)$, $\theta, \gamma > 0$ and integer $r$, there exist constants $c_*, \tau, \rho$ dependent on $\alpha, (\beta_q)_{q \leq r}, \gamma, \theta, r$, such that, if $\sum_q k_q = |\mathrm{supp}(\mathbf{v}_q)| \leq c_* \sqrt{n}$, the Covariance Thresholding algorithm of Table 1 recovers the joint supports of $\mathbf{v}_q$ with high probability.*

*Explicitly, there exists a constant $C > 0$ such that*

$$\mathbb{P}\left\{\widehat{\mathsf{Q}} = \cup_q \mathrm{supp}(\mathbf{v}_q)\right\} \geq 1 - \frac{C}{n^4}. \tag{7}$$

Given the results above, it is useful to pause for a few remarks.

**Remark 2.1.** We focus on a consistent estimation of the joint supports $\cup_q \mathrm{supp}(\mathbf{v}_q)$ of the spikes. In the rank-one case, this obviously corresponds to the standard support recovery. Once this is accomplished, estimating the individual supports poses no additional difficulty: indeed, since $| \cup_q \mathrm{supp}(\mathbf{v}_q))| = O(\sqrt{n})$ an extra step with $n$ fresh samples $\mathbf{x}_i$ restricted to $\widehat{\mathsf{Q}}$ yields estimates for $\mathbf{v}_q$

with $\ell_2$ error of order $\sqrt{k/n}$. This implies consistent estimation of $\mathrm{supp}(\mathbf{v}_q)$ when $\mathbf{v}_q$ have entries bounded below as in Assumption A2.

**Remark 2.2.** Recovering the signed supports $\mathsf{Q}_{q,+} = \{i \in [p] : v_{q,i} > 0\}$ and $\mathsf{Q}_{q,-} = \{i \in [p] : v_{q,i} < 0\}$ is possible using the same technique as recovering the supports $\mathrm{supp}(\mathbf{v}_q)$ above, and poses no additional difficulty.

**Remark 2.3.** Assumption A2 requires $|v_{q,i}| \geq \theta/\sqrt{k_q}$ for all $i \in \mathsf{Q}_q$. This is a standard requirement in the support recovery literature [Wai09, MB06]. The second part of assumption A2 guarantees that when the supports of two spikes overlap, their entries are roughly of the same order. This is necessary for our proof technique to go through. Avoiding such an assumption altogether remains an open question.

Our covariance thresholding algorithm assumes knowledge of the correct support sizes $k_q$. Notice that the same assumption is made in earlier theoretical, e.g. in the analysis of SDP-based reconstruction by Amini and Wainwright [AW09]. While this assumption is not realistic in applications, it helps to focus our exposition on the most challenging aspects of the problem. Further, a ballpark estimate of $k_q$ (indeed $\sum_q k_q$) is actually sufficient, with which we use the following steps in lieu of Steps 7, 8 of Algorithm 1.

    7: Define $\mathbf{s}'_q \in \mathbb{R}^p$ by

$$s'_{q,i} = \begin{cases} \widehat{v}_{q,i} & \text{if } |\widehat{v}_{q,i}| > \theta/(2\sqrt{k_0}) \\ 0 & \text{otherwise.} \end{cases} \tag{8}$$

    8: Return

$$\widehat{\mathsf{Q}} = \cup_q \{i \in [p] : |(\widehat{\mathbf{\Sigma}}' \mathbf{s}'_q)_i| \geq \rho \}. \tag{9}$$

The next theorem shows that this procedure is effective even if $k_0$ overestimates $\sum_q k_q$ by an order of magnitude. Its proof is deferred to Section 2.

**Theorem 3.** *Consider the spiked covariance model of Eq. (1). For any $\alpha, \beta \in (0, \infty)$, let constants $c_*, \tau, \rho$ be given as in Theorem 2. Further assume $k = \sum_q |\mathrm{supp}(\mathbf{v}_q)| \leq c_* \sqrt{n}$, and $\sum_q k \leq k_0 \leq 20 \sum_q k_q$. Then, the Covariance Thresholding algorithm of Table 1, with the definitions in Eqs. (8) and (9), recovers the joint supports of $\mathbf{v}_q$ successfully, i.e.*

$$\mathbb{P}\left(\widehat{\mathsf{Q}} = \cup_q \mathrm{supp}(\mathbf{v}_q)\right) \geq 1 - \frac{C}{n^4}. \tag{10}$$

## 3 Practical aspects and empirical results

Specializing to the rank one case, Theorems 1 and 2 show that Covariance Thresholding succeeds with high probability for a number of samples $n \gtrsim k^2$, while Diagonal Thresholding requires $n \gtrsim k^2 \log p$. The reader might wonder whether eliminating the $\log p$ factor has any practical relevance or is a purely conceptual improvement. Figure 1 presents simulations on synthetic data under the strictly sparse model, and the Covariance Thresholding algorithm of Table 1, used in the proof of Theorem 2. The objective is to check whether the $\log p$ factor has an impact at moderate $p$. We compare this with Diagonal Thresholding.

We plot the empirical success probability as a function of $k/\sqrt{n}$ for several values of $p$, with $p = n$. The empirical success probability was computed by using 100 independent instances of the problem. A few observations are of interest: $(i)$ Covariance Thresholding appears to have a significantly larger success probability in the 'difficult' regime where Diagonal Thresholding starts to fail; $(ii)$ The curves for Diagonal Thresholding appear to decrease monotonically with $p$ indicating that $k$ proportional to $\sqrt{n}$ is not the right scaling for this algorithm (as is known from theory); $(iii)$ In contrast, the curves for Covariance Thresholding become steeper for larger $p$, and, in particular, the success probability increases with $p$ for $k \leq 1.1\sqrt{n}$. This indicates a sharp threshold for $k = \text{const} \cdot \sqrt{n}$, as suggested by our theory.

In terms of practical applicability, our algorithm in Table 1 has the shortcomings of requiring knowledge of problem parameters $\beta_q, r, k_q$. Furthermore, the thresholds $\rho, \tau$ suggested by theory need not

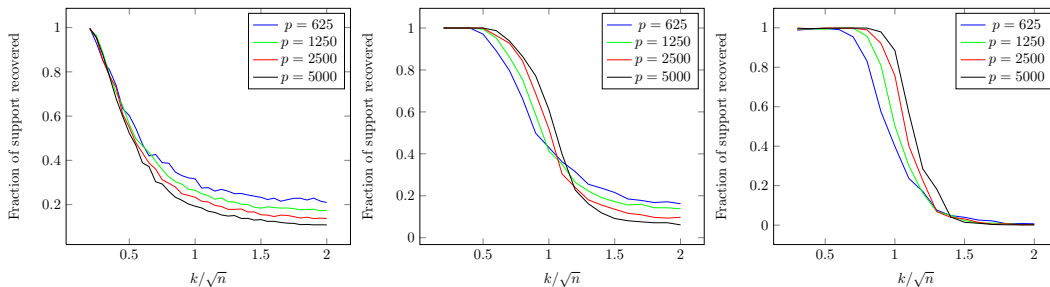

Figure 1: The support recovery phase transitions for Diagonal Thresholding (left) and Covariance Thresholding (center) and the data-driven version of Section 3 (right). For Covariance Threshold­ing, the fraction of support recovered correctly *increases* monotonically with $p$, as long as $k \leq c\sqrt{n}$ with $c \approx 1.1$. Further, it appears to converge to one throughout this region. For Diagonal Thresh­olding, the fraction of support recovered correctly *decreases* monotonically with $p$ for all $k$ of order $\sqrt{n}$. This confirms that Covariance Thresholding (with or without knowledge of the support size $k$) succeeds with high probability for $k \leq c\sqrt{n}$, while Diagonal Thresholding requires a significantly sparser principal component.

be optimal. We next describe a principled approach to estimating (where possible) the parameters of interest and running the algorithm in a purely data-dependent manner. Assume the following model, for $i \in [n]$

$$\mathbf{x}_i = \boldsymbol{\mu} + \sum_q \sqrt{\beta_q} u_{q,i} \mathbf{v}_q + \sigma \mathbf{z}_i,$$

where $\boldsymbol{\mu} \in \mathbb{R}^p$ is a fixed mean vector, $u_i$ have mean 0 and variance 1, and $\mathbf{z}_i$ have mean 0 and co­variance $\mathrm{I}_p$. Note that our focus in this section is not on rigorous analysis, but instead to demonstrate a principled approach to applying covariance thresholding in practice. We proceed as follows:

**Estimating $\boldsymbol{\mu}, \sigma$:** We let $\widehat{\boldsymbol{\mu}} = \sum_{i=1}^n \mathbf{x}_i/n$ be the empirical mean estimate for $\mu$. Further letting $\overline{\mathbf{X}} = \mathbf{X} - \mathbf{1}\widehat{\boldsymbol{\mu}}^\mathsf{T}$ we see that $pn - (\sum_q k_q)n \approx pn$ entries of $\overline{\mathbf{X}}$ are mean 0 and variance $\sigma^2$. We let $\widehat{\sigma} = \mathrm{MAD}(\overline{\mathbf{X}})/\nu$ where $\mathrm{MAD}(\cdot)$ denotes the median absolute deviation of the entries of the matrix in the argument, and $\nu$ is a constant scale factor. Guided by the Gaussian case, we take $\nu = \Phi^{-1}(3/4) \approx 0.6745$.

**Choosing $\tau$:** Although in the statement of the theorem, our choice of $\tau$ depends on the SNR $\beta/\sigma^2$, we believe this is an artifact of our analysis. Indeed it is reasonable to threshold 'at the noise level', as follows. The noise component of entry $i,j$ of the sample covari­ance (ignoring lower order terms) is given by $\sigma^2 \langle \mathbf{z}_i, \mathbf{z}_j \rangle/n$. By the central limit theo­rem, $\langle \mathbf{z}_i, \mathbf{z}_j \rangle/\sqrt{n} \overset{\mathrm{d}}{\Rightarrow} \mathsf{N}(0,1)$. Consequently, $\sigma^2 \langle \mathbf{z}_i, \mathbf{z}_j \rangle/n \approx \mathsf{N}(0, \sigma^4/n)$, and we need to choose the (rescaled) threshold proportional to $\sqrt{\sigma^4} = \sigma^2$. Using previous estimates, we let $\tau = \nu' \cdot \widehat{\sigma}^2$ for a constant $\nu'$. In simulations, a choice $3 \lesssim \nu' \lesssim 4$ appears to work well.

**Estimating $r$:** We define $\widehat{\boldsymbol{\Sigma}} = \overline{\mathbf{X}}^\mathsf{T}\overline{\mathbf{X}}/n - \sigma^2\mathrm{I}_p$ and soft threshold it to get $\eta(\widehat{\boldsymbol{\Sigma}})$ using $\tau$ as above. Our proof of Theorem 1 relies on the fact that $\eta(\widehat{\boldsymbol{\Sigma}})$ has $r$ eigenvalues that are separated from the bulk of the spectrum[4]. Hence, we estimate $r$ using $\widehat{r}$: the number of eigenvalues separated from the bulk in $\eta(\widehat{\boldsymbol{\Sigma}})$.

**Estimating $\mathbf{v}_q$:** Let $\widehat{\mathbf{v}}_q$ denote the $q^{\text{th}}$ eigenvector of $\eta(\widehat{\boldsymbol{\Sigma}})$. Our theoretical analysis indicates that $\widehat{\mathbf{v}}_q$ is expected to be close to $\mathbf{v}_q$. In order to denoise $\widehat{\mathbf{v}}_q$, we assume $\widehat{\mathbf{v}}_q \approx (1-\delta)\mathbf{v}_q + \boldsymbol{\varepsilon}_q$, where $\boldsymbol{\varepsilon}_q$ is additive random noise. We then threshold $\mathbf{v}_q$ 'at the noise level' to re­cover a better estimate of $\mathbf{v}_q$. To do this, we estimate the standard deviation of the "noise" $\boldsymbol{\varepsilon}$ by $\widehat{\sigma_{\boldsymbol{\varepsilon}}} = \mathrm{MAD}(\mathbf{v}_q)/\nu$. Here we set –again guided by the Gaussian heuristic– $\nu \approx 0.6745$. Since $\mathbf{v}_q$ is sparse, this procedure returns a good estimate for the size of the noise deviation. We let $\eta_H(\widehat{\mathbf{v}}_q)$ denote the vector obtained by hard thresholding $\widehat{\mathbf{v}}_q$: set

$(\eta_H(\widehat{\mathbf{v}}_q))_i = \widehat{\mathbf{v}}_{q,i}$ if $|\widehat{v}_{q,i}| \geq \nu' \widehat{\sigma_{\boldsymbol{\varepsilon}_q}}$ and 0 otherwise. We then let $\widehat{\mathbf{v}}_q^* = \eta(\widehat{\mathbf{v}}_q)/\|\eta(\widehat{\mathbf{v}}_q)\|$ and return $\widehat{\mathbf{v}}_q^*$ as our estimate for $\mathbf{v}_q$.

Note that –while different in several respects– this empirical approach shares the same philosophy of the algorithm in Table 1. On the other hand, the data-driven algorithm presented in this section is less straightforward to analyze, a task that we defer to future work.

Figure 1 also shows results of a support recovery experiment using the 'data-driven' version of this section. Covariance thresholding in this form also appears to work for supports of size $k \leq$ const$\sqrt{n}$. Figure 2 shows the performance of vanilla PCA, Diagonal Thresholding and Covariance Thresholding on the "Three Peak" example of Johnstone and Lu [JL04]. This signal is sparse in the wavelet domain and the simulations employ the data-driven version of covariance thresholding. A similar experiment with the "box" example of Johnstone and Lu is provided in the supplement. These experiments demonstrate that, while for large values of $n$ both Diagonal Thresholding and Covariance Thresholding perform well, the latter appears superior for smaller values of $n$.

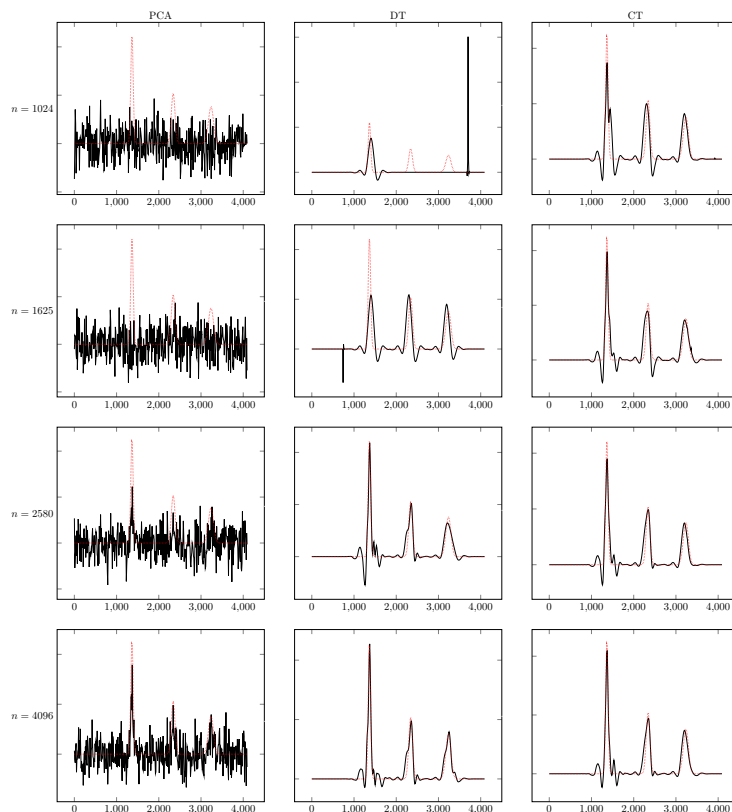

Figure 2: The results of Simple PCA, Diagonal Thresholding and Covariance Thresholding (respectively) for the "Three Peak" example of Johnstone and Lu [JL09] (see Figure 1 of the paper). The signal is sparse in the 'Symmlet 8' basis. We use $\beta = 1.4, p = 4096$, and the rows correspond to sample sizes $n = 1024, 1625, 2580, 4096$ respectively. Parameters for Covariance Thresholding are chosen as in Section 3, with $\nu' = 4.5$. Parameters for Diagonal Thresholding are from [JL09]. On each curve, we superpose the clean signal (dotted).

## Footnotes

[1]Throughout the paper, we follow the convention of denoting scalars by lowercase, vectors by lowercase boldface, and matrices by uppercase boldface letters.

[2]Throughout the introduction, we write $f(n) \gtrsim g(n)$ as a shorthand of '$f(n) \geq C \, g(n)$ *for a some constant* $C = C(\beta, \alpha)$'. Further $C$ denotes a constant that may change from point to point.

[3] Note that [CS12] also provide a finite $n$ bound for the spectral norm of $f(\mathbf{Z}^{\mathsf{T}}\mathbf{Z}/n)$ via the moment method, but this bound diverges with $n$ and does not give a result of the type of Eq. (5).

[4]The support of the bulk spectrum can be computed numerically from the results of [CS12].

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
