[Reviews · NeurIPS 2014]

Submitted by Assigned_Reviewer_42

Note: the authors have violated the formatting rules by largely widening the margins of the text. I find this practice irrespecutful to other authors who have respected the rules to convey their message. Thus, the manuscript, as it is, should not be accepted for publication.

The paper is well written and technically well executed. The main contributions are Theorems 1 and 2.

In Theorem 1, the authors bound the estimation error in the principal components of the covariance thresholding algorithm of Krauthgamer et al. The proof uses standard bounding techniques and covering numbers on a clever decomposition of the random matrix under study.

Theorem 2 concerns with proving that the second phase of the covariance thresholding algorithm recovers the sparse support of the principal components consistently.

The authors propose heuristics to choose the constants appearing in the theorems. However, results are only validated on one synthetic dataset on which differences in performance between methods become statistically indistinguishable for rather moderate sample sizes (n=2000).

In general, a wider empirical validation of the proposed techniques would undoubtedly add value to the submission. This is specially important given the low interpretability of the constants involved in the theoretical analysis and the need to resort to a collection of heuristics to properly set the problem parameters, i.e., support size, signal-to-noise ratio, et cetera.
Summary: Presentation of consistency results for the sparse principal component analysis algorithm of Krauthgamer et al. Theoretical analysis is well executed but empirical validation could be more exhaustive.

The paper violates the conference's formatting rule because of smaller margins.

Submitted by Assigned_Reviewer_43

The paper studies sparse PCA via a covariance thresholding algorithm (originally proposed by Johnston and Lu, recently modified by Krauthgamer, Nadler and Vilenchik) and provides new tighter analysis. The sparse PCA problem is studied under the spiked covariance model. The assumptions made for analysis are standard in this literature. However, the analysis assumes knowledge of key parameters of the observation model which is undesirable but authors point out how to estimate those parameters in a data driven manner and confirm with empirical results the utility of those estimates. The empirical results are satisfactory. The paper is very clearly written and easy to follow; authors do a good job of sketching out the current state of research in sparse PCA and put their contributions in context; some minor typos and grammatical errors so a proof read is recommended before the final version. The paper scores high on originality and significance.
Summary: The paper studies sparse PCA via a covariance thresholding algorithm (originally proposed by Johnston and Lu, recently modified by Krauthgamer, Nadler and Vilenchik) and provides new tighter analysis. Overall, a good paper.

Submitted by Assigned_Reviewer_44

The focus of this paper is on a theoretical analysis of sparse PCA under a spiked covariance model, where the leading eigenvectors are assumed to be L_0 sparse.

Quality & Significance: The key contribution is a proof that a simple and computationally efficient covariance thresholding algorithm, suggested by Krauthgamer et. al., indeed performs better than simple diagonal thresholding
and can recover sparsity patterns with a sparsity of at most O(sqrt(n)).

It thus makes a nice contribution to the literature on the theoretical analysis of sparse PCA algorithms. The ideas presented may also be applicable to other sparse settings and problems.

Clarity and Originality: Overall, the paper is reasonably well written, and the result is new and interesting. The proof appears in the supplementary material
and has some quite original parts. It is based on a clever decomposition of the matrix into signal and noise parts coupled with epsilon-net and concentration of measure results as well as splitting of the data into two parts one used for initial estimate, the second for refinement. I did not fully verify its correctness.

Unfortunately, there are quite a few typos, unclear sentences and at times inconsistent notation in both paper and supplementary.

One small remark about the simulation results - while I appreciate that the focus of the paper is on the theoretical result, figure 3 and similar one in supplementary are not very informative. What is the main message here and what do we learn from these ?

One page 8, in description of data driven algorithm. Assuming sigma \neq 1, there seem to be a few sigma^2 missing, both in "Consequently, (z,z_j)/n ~ N(0,sigma^4/n) and later on in \hat\Sigma = \bar X^T \bar X/n - I_p (probably should be \hat sigma^2 I_p) ?

Some further comments:

* abstract - sentence "Recent conditional lower bounds..." is rather unclear.

* Consider explicitly stating that k is known already on page 2, rather than on page 3.

* section 2 - I guess r = rank is also an input parameter of the algorithm ?
Also, not very clear what is the output of the algorithm. It seems like a set of indices, while the algorithm is called covariance thresholding, and the problem is sparse PCA...

* the exposition and flow of the paper can be improved, in particular some unclear and disconnected sentences at top of page 4 and sharp transition at top of page 5.

* While I understand that Eq. 3 and 4 are "intuitive" I still don't understand in what sense is Eq. 4 approximate, since thresholding is not an additive operation, namely eta(a+b) \neq eta(a) + eta(b)

* in proof of theorem 1, supplementary, eq. 4 second line should it not be v_q (v_q')^T ? Also, you seem to use Q^q instead of previous Q_q. Also, what is Q^c (where was it defined) ?

typos: too many to mention all, but
in abstract - why ) after sqrt(n) ?
page 3 answer positively answer
why lower bar theta in page 5 and is this different from theta in page 2 ?
in statement of theorem 3, Eqs. (8) and (8) ??? also sum_q k should probably be
sum_q k_q ?

In summary, if accepted, authors should do a thorough reading of paper and supplementary to make paper and proofs more readable.
Summary: This paper presents a theoretical analysis that a covariance thresholding algorithm
can solve an L_0 sparse PCA problem better than simple diagonal thresholding.
It advances the understanding of the statistical vs. computational difficulty
of sparse PCA.

Submitted by Assigned_Reviewer_45

I think the paper is well-written (modulo initial typos) and suitable for publication. I have nothing further to add to my previous review.
Summary: I think the paper is well-written (modulo initial typos) and suitable for publication. I have nothing further to add to my previous review.

Submitted by Meta_Reviewer_9

The authors affirm a conjecture posed by Krauthgamer, Nadler and Vilenchik. regarding the performance of an algorithm for sparse PCA. Namely, they consider doing sparse PCA by (essentially) performing PCA on a thresholded sample covariance matrix. They show that the algorithm succeeds in recovering the joint support of the sparse components up to the threshold k = O(\sqrt(n)), where "k" is the sparsity index and "n" the sample size. This is believed to be the threshold achievable by polynomial-time algorithms. The result closes the gap k \in [\sqrt{n/log p}, \sqrt{n}] by showing an efficient algorithm that works in this regime.

I find the results interesting and the paper well-written, modulo some typos and inconsistencies which I assume are introduced from translating a longer version to NIPS format.

I have a few concerns and questions:
(1) The authors focus on recovering the union of the supports of the components in the multi-spiked case. First, this is not clear from the start until presented in the theorems and discussed in Remark 2.1. It is worth emphasizing this early on. I also don't quite agree with the content of Remark 2.1. The authors mention that given the joint support and "n" fresh samples, it is possible to consistently estimate the individual supports, because one can get consistent estimates of the individual components (v_q) once one restrict to the joint support. I agree that one can get consistency for v_q's by usual PCA restricted to joint support, but this is \ell_2 consistency. It is not clear whether thresholding these eigenvectors will give the correct supports. In particular, the problem seems to depend on the relative sizes of the supports and the number of components "r". (Consider many components having very small and very large supports).

(2) The second half of assumption (A2) is a bit questionable as the authors point out later in Remark 2.3. It is better to move that remark closer to where the assumption is presented. I have no serious objection here, as the results are interesting even in the single component case. However, I want to point out that by symmetry (switching q and q'), the second half of assumption is effectively assuming |v_{q,i}/v_{q',i}| = \gamma. That is, the reverse inequality also holds by assumption.

(3) Is the passage of the estimated eigenvector once more through the estimated covariance matrix really necessary? This is step (7) of Algorithm 1. What seems to be implied is that without it consistency is not achieved. In particular, merely thresholding the eigenvectors does not provide the correct supports. Is this true or is the analysis inconclusive in this regard? Any comments clarifying this issue is helpful. Also, you might point out that you are looking at a slightly modified version of what was proposed by Krauthgamer, Nadler and Vilenchik.

Some minor issues:
- Algorithm (1) appears before some of the quantities involved are introduced. It is better to move it further down.

- p.3, in step 3 of rough algorithm, B should be defined in terms of \hat{v}_1 and not the population version v_1.

- Algorithm 1, definition of G': the upper limit of the sum should be 2n.

- p.4, Table 1 should be replaced with Algorithm 1.

- p.4, ... provides bounds "on" the estimation error.

- p.5, ... "a" kernel inner product random matrix.

- p.5, in the first displayed equation \tilde{z}_i is not defined.
Summary: I think the paper is well-written and the results are interesting contributions to sparse PCA literature. Some might question the relatively small gain of log(p), but I think it is interesting from a theoretical point of view. The authors are also convincing in motivating the gain practically.
Author Feedback
Author rebuttal: We thank the referees for their comments, and address below their concerns.

In the shortened version (addressing Assigned_Reviewer_42's observation below)
we have only made the following changes:
- Reduced size of Figure 2
- Corrected typos mentioned by Assigned_Reviewer_44

It is available at:
sparsepcacovthr.wordpress.com

If accepted, the final version will incorporate changes addressing
*all other concerns* as mentioned below.

%%%%%%%%%%%%%%%%%%%%%%%%%%%%%%%%%%%%%%%%%%%%%%%%%%%%%%

Assigned_Reviewer_42:

We thank the reviewer for her/his feedback.
Firstly, we sincerely apologize for the incorrect formatting. It was largely due to a
typesetting error and has been corrected. **The corrected version fits the page limit.**

We agree that a wider empirical validation would add value, and it is
indeed a part of our future plan.
Our focus was to provide a polynomial-time algorithm that saturates the O(\sqrt{n})
support recovery limit. Notice that there is *no other algorithm* that
provably achieves the same at the moment.

- " The authors propose heuristics to choose the constants appearing in the
theorems. However, results are only validated on one synthetic dataset on
which differences in performance between methods become statistically
indistinguishable for rather moderate sample sizes (n=2000)."

In Figure 2 we fix the signal dimension p and increase the number of
samples n. When n is large enough (n>2500), both covariance thresholding and
diagonal thresholding work well. This is expected: with a large amount
of data, most algorithms succeed. However, for moderate number of
samples (n=1024, 1625) covariance thresholding outperform diagonal
thresholding.This gap is for a relatively small signal (p=4096): our theory
establishes that the gap increases as p becomes larger.

Further Figure 2, along with the similar figure in the supplement, demonstrate
resilience of our method to modeling assumptions: exact sparsity, A1 and A2.

Finally, we used this synthetic data because both of these examples were also
used in Johnstone and Lu's original paper on sparse PCA (introducing diagonal
thresholding). Hence, they provide a natural benchmark.

%%%%%%%%%%%%%%%%%%%%%%%%%%%%%%%%%%%%%%%%%%%%%%%%%%%%%%

Assigned_Reviewer_43:

We thank the reviewer for her/his comments.

%%%%%%%%%%%%%%%%%%%%%%%%%%%%%%%%%%%%%%%%%%%%%%%%%%%%%%

Assigned_Reviewer_44:

We thank the reviewer for her/his comments.
We have re-read the submission and corrected all typos that we came across (including, of course, those
mentioned by the reviewer). Below are answers to specific questions.

- "while I appreciate that the focus of the paper is on the theoretical result,
figure 3 and similar one in supplementary are not very informative.
What is the main message here and what do we learn from these ? "

The objective of the simulations in Figure 2 is to show resilience of the
method to relaxing of exact sparsity and assumptions A1, A2 of the paper.

- "One page 8, in description of data driven algorithm. Assuming sigma \neq 1,
there seem to be a few sigma^2 missing, both in "Consequently, (z,z_j)/n ~
N(0,sigma^4/n) and later on in \hat\Sigma = \bar X^T \bar X/n - I_p
(probably should be \hat sigma^2 I_p) ?"

We agree and have corrected that portion, thanks.

- "abstract - sentence "Recent conditional lower bounds..." is rather unclear."

The relevant paper is cited in abstract, and the point is elucidated further in
introduction.

- "section 2 - I guess r = rank is also an input parameter of the algorithm ?"

r is defined to be the number of spikes. For the proof we require it known,
but we will provide a heuristic estimation procedure in the practical aspects
section.

- "Also, not very clear what is the output of the algorithm. It seems like a
set of indices, while the algorithm is called covariance
thresholding, and the problem is sparse PCA..."

The sparse PCA task we consider is of support recovery under the statistical
spiked covariance model. Other error metrics are studied in the literature but we do not address
them. However, once the support is correctly identified, classical
estimators
of the covariance and its principal component can be applied, by
restricting the data to the identified support.

- "the exposition and flow of the paper can be improved, in particular some unclear
and disconnected sentences at top of page 4 and sharp transition at
top of page 5. "

The mentioned portions are edited to improve readability.

-"While I understand that Eq. 3 and 4 are "intuitive" I still don't understand in
what sense is Eq. 4 approximate, since thresholding is not an
additive operation, namely eta(a+b) \neq eta(a) + eta(b)"

We completely agree: indeed the nonlinearity of \eta( ) is the main
technical challenge in the proof.
The proof or Theorem 1 shows in what sense Eq. 4 holds approximately
(namely, in operator norm). We will explicitly state this in the paper.

- "in proof of theorem 1, supplementary, eq. 4 second line should it not be
v_q (v_q')^T ? Also, you seem to use Q^q instead of previous Q_q. Also,
what is Q^c (where was it defined) ?"

Corrected, thanks. Q is defined to be the union of supports of v_q. Q^c
is its complement. The complement notation is consistent throughout, and
we added a sentence about it in the notation section.

- "why lower bar theta in page 5 and is this different from theta in page 2 ?"

That is a typo. We have corrected it, thanks.